# Food Safety Knowledge among Professional Food Handlers in Slovenia: The Results of a Nation-Wide Survey

**DOI:** 10.3390/foods12142735

**Published:** 2023-07-18

**Authors:** Mojca Jevšnik, Andrej Kirbiš, Stanka Vadnjal, Urška Jamnikar-Ciglenečki, Andrej Ovca, Matic Kavčič

**Affiliations:** 1Faculty of Health Sciences, University of Ljubljana, Zdravstvena pot 5, 1000 Ljubljana, Slovenia; andrej.ovca@zf.uni-lj.si (A.O.); matic.kavcic@zf.uni-lj.si (M.K.); 2Institute of Food Safety, Feed and Environment, Veterinary Faculty, University of Ljubljana, Gerbičeva 60, 1000 Ljubljana, Slovenia; andrej.kirbis@vf.uni-lj.si (A.K.); stanka.vadnjal@vf.uni-lj.si (S.V.); urska.jamnikar@vf.uni-lj.si (U.J.-C.)

**Keywords:** food safety, food handler, knowledge, hygiene

## Abstract

The authors present and discuss the results of a nation-wide survey on food safety knowledge among professional food handlers in Slovenia. The data were collected via a telephone survey using a well-established questionnaire adapted to the Slovenian context. Altogether, 601 respondents from hotels, restaurants, catering, and confectionery units completed the questionnaire. To assess food safety knowledge among food handlers in both general and specific domains, three indexes (a General Knowledge Index, a Personal Knowledge Index, and a Temperature Knowledge Index) were created. Among them, the Temperature Knowledge Index revealed the largest gaps in knowledge. An insufficient transfer of food safety knowledge from managers and chefs to assistant chefs and kitchen assistants in establishments where more persons handle food was evident, while a course titled “Hygiene Minimum” of standardised training from the past still significantly contributes to food safety knowledge. The results suggest a need for improvement in the current system of food safety training courses for professional food handlers in Slovenia. The human factor in the food supply chain still has a significant role in ensuring food safety culture, and therefore must become a more important part of the food safety management system.

## 1. Introduction

A Slovenian survey on people’s eating habits showed that 57.5% (N = 393; 29.2%; the percentage of the sample; N = 1,302,132; 78.2%; the percentage of the population) of the adult population (aged 18–64) “eats out” during the week (at work, on the go, in a bar or restaurant) [1]. Many authors [2,3] pointed out that food service establishments are considered to be a significant source of foodborne diseases (FBD) (e.g., campylobacteriosis, and salmonellosis), something which has also been confirmed by European Food Safety Authority (EFSA) data [4] from the pre-COVID period. A total of 28.1% of the foodborne outbreaks (FBO) in European Union (EU) countries [4] occurred as a result of eating foods at various food service establishments.

Food handlers working at retail establishments, in food services, or in catering were identified as a frequent source of FBD [2,5]. Previous observational surveys [6,7,8,9,10] found that professional food handlers’ most frequent errors were made in connection with bare-hand contact with food, improper hand-washing practices, and insufficient cleaning of processing equipment, all of which resulted in subsequent potential outbreaks. Smigic et al. [11] identified the largest knowledge gaps in relation to temperature control, the sources of contamination of foods and foodstuffs, and high-risk foods for food handlers from three different European countries. The causes include the insufficient food safety training of food handlers, the insufficient knowledge of food handlers, negligence, and/or inefficient supervision by supervisors [8,12,13,14].

Food business operators have to ensure that all their staff engaged in food-handling activities are both suitably trained and/or instructed in food hygiene, and also able to maintain a high level of personal hygiene [15,16]. However, regular food safety training does not always encourage the appropriate attitudes and behaviour among food handlers in terms of their hygiene practices [11,17,18]. Yu et al. [19] pointed out that behaviour-based training was indeed effective in changing hygiene behaviour over time. Several studies [17,20,21] have also established that the theoretical training of food handlers is not always connected to their own views and attitudes to food safety, or their own daily practices. Soon et al. [22] found that periodical training is needed to obtain and maintain the desired knowledge. Others have also pointed out that the kind of training that changes the level of knowledge does not always translate into changes in food-handling practices [7,23,24,25], as translating knowledge into practice is a complex process [26]. Clayton et al. [24] report that food handlers were aware of the food safety behaviours they should be carrying out, but 63% of them admitted that they did not always do so. However, food handlers reported performing food safety practices, especially hand-washing, much more frequently than they did in their daily practice [12,25,27,28].

From a food handler’s perspective, training provided by experts in a company or by their supervisors during their daily work is the most effective, especially for those in food production [8], although it was established that food businesses—especially small and medium enterprises (SMEs)—often do not have either satisfactory training practices or policies for their staff. The latter is crucial because European education and training systems often do not sufficiently provide the right skills for employability, and do not sufficiently engage with businesses or employers in order to bring the learning experience closer to the realities of the world of work [29].

Prior to the European food safety legislation coming into force, Slovenia abandoned the special standardised food hygiene educational programme titled the “Hygiene Minimum”, which was mandatory at the national level until 2004 for all professional food handlers before commencing work, and which had to be renewed every five years [30]. However, when food business operators ignore the continuous training of their employees in order to reduce costs, a food handler’s level of knowledge often depends on their previous formal education and his or her self-initiative. This has also been confirmed in a study by Jevšnik et al. [3], who estimated that 5.8% of respondents who work in catering establishments have never taken part in food safety training courses.

Jevšnik et al. [31] emphasised that food safety training and individual awareness were the most important tools for food safety assurance, and this means that every food handler requires complex and individual personal management. The human factor must be equally discussed among all the other risk factors. For food safety, it is essential that every person in the food supply chain understands and fulfils their responsibilities, and that they rely upon the previous and next steps in the chain.

Various studies have been conducted to identify which psychosocial and organisational factors are the most important determinants for the safe food-handling behaviours of food handlers working at food service establishments [24,32,33,34]. However, most studies in this area have traditionally focused on knowledge and attitude as the two key constructs influencing food handlers’ behaviours [24].

The aim of this study was to detect the most important gaps in food safety knowledge among professional food handlers in catering establishments in Slovenia, and to compare the results with a similar Austrian study [11]. Furthermore, the study intends to elucidate any relationships among different food establishment and food handler characteristics, such as age, gender, education, work experience, previous food safety training, and food safety knowledge. In addition, we also wanted to evaluate the role of the above mentioned “Hygiene Minimum” training course. The results of this study can help to improve current food safety educational materials, and provide a foundation for the design of new training models for food handlers in catering establishments.

## 2. Materials and Methods

### 2.1. The Sample and Procedures

A nationwide, cross-sectional telephone survey was designed to investigate food safety knowledge among food handlers in food service establishments in Slovenia. A simple random sample was executed on the level of food service establishments for the selection of respondents. For the sampling frame, we used a list of all food establishments registered by the Chamber of Craft and Small Business of Slovenia. All 4300 food establishments were selected for the study. A total of 601 establishments agreed to participate, and provided a respondent to answer the telephone survey questions (with a response rate of 14%). A comparison of the realised sample according to region, type, and size of establishment against the registered food establishments did not show any significant discrepancies. The respondent within the food establishment was chosen at random. First the interviewer asked for the total number of employees who handle food in the establishment, and this was followed by writing down the employees’ names and functions. Finally, the computer program randomly selected a person to participate in the survey. The collection of data was carried out by a professional research agency that specialises in computer-assisted telephone interviewing, and this took place in the period from May to June of 2018. No special reward was offered for participation, with the exception of the opportunity to receive a leaflet on food safety via e-mail afterwards. Respondents were assured that their participation would be confidential and voluntary. All subjects gave their informed consent for inclusion before they participated in the study. The study was conducted in accordance with the Declaration of Helsinki, and the protocol was approved by Administration of the Republic of Slovenia for Food Safety, Veterinary Sector and Plant Protection and Veterinary Faculty University of Ljubljana (No. P4-0092).

### 2.2. The Questionnaire and Measures

The questionnaire used in the survey was adapted with permission from Pichler et al. [11], and was previously developed and used by Dworkin et al. [35] and Panchal et al. [36]. It was translated into Slovenian and adjusted for a Slovenian context. The questionnaire was then tested for content validity and piloted among six food safety experts and ten professional food handlers in catering establishments to determine the clarity of the questions/statements, identify additional response options, and estimate the time required to complete the survey. The questionnaire was revised based on a pre-test, and some modifications were made regarding work satisfaction, food leftovers, and waste.

The questionnaire consisted of six sections and contained 56 questions on the following topics: time and temperature in food safety management; cross-contamination; hand hygiene; cleaning, disinfection, and employee health and personal hygiene; job satisfaction; food leftovers; waste. Additional information was also collected on the general and socio-demographic characteristics of the study participants (age, education level, length of employment, work experience, and previous food hygiene training).

In order to assess the food safety knowledge of the food handlers in both general and specific domains, we composed three indexes. First, there was the Temperature Knowledge Index (TKI), which consisted of six items (questions) on appropriate storing, cooling, and heating temperatures. Responses were recoded so that “0” indicated an incorrect answer (or, “I do not know,”), and “1” indicated a correct answer. The TKI was thus measured in a range from 0 to 6 points in total, where 0 meant that the respondent did not answer any of the questions correctly, and 6 meant that the respondent answered all of the questions correctly. The TKI descriptive statistics were as follows: n = 601, M = 2.27, Std. dev. = 1.08, Min = 0, Max = 5, skew = 0.96, kurt = 0.199.

Second, the Personal Hygiene Index (PHI) consisted of eight items (questions) on hand-washing. Responses were recoded so that “0” indicated an incorrect answer (or, “I do not know,”), and “1” indicated a correct answer. The PHI was measured in a range from 0 to 8 points in total, where 0 meant that the respondent did not answer any of the questions correctly, while 8 meant that the respondent answered all of the questions correctly. The PHI descriptive statistics were as follows: n = 601, M = 6.86, Std. dev. = 0.90, Min = 3, Max = 8, skew = −1.93, kurt = 2.62.

Finally, the third index, the General Knowledge Index (GKI), measured food handling and general hygiene knowledge in the kitchen. It consisted of 45 items (questions) that had been previously recoded so that “0” indicated an incorrect answer (or, “I do not know,”), and “1” indicated a correct answer. The values of the index ranged from 0 to 45 points, where 0 meant that the respondent did not answer any of the questions correctly, and 45 meant that the respondent answered all of the questions correctly. The GKI descriptive statistics were as follows: n = 601 M = 35.60, Std. dev. = 3.10, Min = 10, Max = 42, skew = −2.20, kurt = 11.11.3.

### 2.3. Data Analysis

A quantitative data analysis was conducted using the software IBM SPSS Statistics 22. In addition to descriptive statistics, we also made use of a various bivariate analysis, a chi-square test, logistic regression, a one-way analysis of variance (ANOVA), an independent samples t-test, and a Pearson and Spearman correlation coefficient. After testing for violations of assumptions, a multiple linear regression analysis for food safety knowledge (GKI) was carried out. The statistical significance of this study was set at *p* < 0.05.

## 3. Results and Discussion

### 3.1. Sample Characteristics

The study involved a total of 601 food handlers, 51.3% of whom were men. A total of 41.3% of respondents had a form of professional secondary technical education (for cooks and bartenders), whereas 28.3% of respondents had another form of secondary technical education. Most of the respondents (86.1%) had Slovenian as their native language, and 38.4% of respondents had a total of 1–10 years of work experience as professional food handlers. In total, 235 of respondents (39.1%) were managers and 206 (34.3%) were chefs. More than half of the respondents (57.4%) had attended the previously mentioned Slovenian “Hygiene Minimum” training course, while 94.2% had attended one of the current food safety training programmes. Table 1 presents the profile of the respondents in the study.

The results regarding the sociodemographic and establishment characteristics associated with individual items are described below.

#### 3.1.1. Gender

We found a slightly smaller, but statistically significant, share (%) of incorrect answers from women in two questions: the possibility of serious diseases if raw beef is not sufficiently heat-treated (Phi = 0.122, *p* = 0.006), and the proper storage of detergents (Phi = 0.96, *p* = 0.019). Conversely, for the question of whether it was correct to collect ice with a glass, the proportion of correct answers was higher from men (Phi = 0.086, *p* = 0.036). For other questions with a higher proportion of incorrect answers, differences according to gender were not statistically significant. Given the weak associations found, gender did not appear to play an important role in this study.

#### 3.1.2. Age

Differences in age were visible for critical indicators (techniques for cooling hot dishes, the labelling and storing of cleaning agents, and collecting ice with a glass), so we performed a logistic regression for these three items. Although older respondents answered questions regarding cooling hot dishes and collecting ice cubes with a glass more correctly and, conversely, younger respondents were less correct in terms of storing detergents in rooms where food was prepared, Nagelkerker’s R^2^ was low in all models. With the first model, we can explain about 1.3% of the variance; with the second model, about 1% of the variance; and with the third model, about 1.8% of the variance. This is quite small, so we concluded that these critical indicators were influenced by other, more decisive factors than age alone.

#### 3.1.3. Education

Regarding the level of education, the analysis does not give a clear picture, and it is interesting to note that those with a high school education (including chefs and waiters) had the worst results when it came to handling ice cubes and collecting ice with a glass. One might suggest that this is due to existing widespread practices, and that it points to a possible discrepancy between knowledge and practice. Not surprisingly, those who had not participated in any food safety training had more knowledge gaps.

Among all respondents (n = 601), there were also 35 (5.8%) who handled food but did not attend any training in the area of food safety. These respondents were statistically more likely to give significantly incorrect answers to questions regarding the proper storage of raw eggs in the refrigerator (Phi = 0.142, *p* = 0.001) and the cooling of hot dishes (Phi = 0.085, *p* = 0.037), and were at the border of statistical significance (Phi = 0.078, *p* = 0.056) in terms of collecting ice with a glass.

#### 3.1.4. Native Language and Country of Birth

Unlike restaurants, where a larger share of food handlers’ native language is Slovenian, in the category of bars, cafes, and confectioneries, and in the category of small food establishments (snack bars), there was a larger share of respondents whose native language was not Slovenian (Cramer’s V = 0.189, *p* < 0.001). In terms of knowledge, contrary to some research from other countries [12], differences with regard to the native language in these analyses did not appear to be statistically significant in Slovenia. Similar results could be seen with respect to the country of birth (a distribution in the types of establishments, Cramer V, 0.140, *p* = 0.006), except for the question of the proper storage of ready meals in cold temperatures (up to 13 °C), where a slightly lower proportion of respondents born in Slovenia answered incorrectly compared to those born elsewhere (Phi = 0.094, *p* = 0.022). The fact that the respondents’ native language did not play a major role could be due to the local socio-cultural context in which food handlers with foreign native languages, especially those from Balkan countries, have a high degree of mutual understanding with others, and this makes training and practices for food handling less difficult.

#### 3.1.5. The “Hygiene Minimum” Training Course

With respect to the formerly mentioned mandatory standardised training course titled “Hygiene Minimum”, it can be said that, in two of the analysed variables, those who completed the training programme turned out slightly better. Among them, there was a statistically significantly higher number of respondents who answered correctly when asked about storing raw eggs in the refrigerator (Phi = 0.101, *p* = 0.013) and collecting ice with a glass (Phi = 0.120, *p* = 0.003).

#### 3.1.6. The Type of Food Establishment

The types of food establishments recorded in the questionnaire were recoded so that restaurants and canteens were combined into one category. Following that, bars, cafes, confectioneries, and snack bars were included into a special category for analysis. The analysis with key variables showed that the type of establishment was statistically and significantly related to knowledge regarding cooling hot dishes (Cramer’s V = 0.137, *p* = 0.007), and that a higher proportion of correct answers was expected among respondents in restaurants and inns. A similar finding was made regarding the storage of raw eggs in the refrigerator (Cramer’s V = 0.149, *p* = 0.003), and the thawing of frozen chicken breasts on a counter (Cramer’s V = 0.136, *p* = 0.008).

#### 3.1.7. Establishment Size

Although the size of the catering establishment was statistically and significantly associated with the most variables, the strength of the association, as elsewhere, was also weak. In accordance with the size of the establishment, the share of incorrect answers to questions regarding improperly stored rice as a source of potential poisoning in humans decreased (Cramer’s V = 0.144, *p* = 0.006), as was the case for the question of the proper storage of raw eggs in a refrigerator (Cramer’s V = 0.155, *p* = 0.002), and the thawing of chicken breasts on a counter (Cramer’s V = 0.159, *p* = 0.002). Interestingly, the reverse was true for answers to the question regarding the proper storage of detergents in food preparation rooms (Cramer’s V = 0.143, *p* = 0.006). The smallest share of incorrect answers regarding cold storage (up to 13 °C) was given by medium-sized establishments (Cramer’s V = 0.145, *p* = 0.006). At the limits of statistical significance (Cramer’s V = 0.113, *p* = 0.052) was the association with the question regarding food safety for consumption and whether its smell and taste was characteristic, in which establishments without tables stood out, with the largest share of incorrect answers (90%). If we were to accept a slightly higher risk, we could also find the following weak association: the proportion of incorrect answers regarding the collection of ice with a glass decreases with the size of the establishment (Cramer’s V = 0.110, *p* = 0.063).

### 3.2. The Factors Associated with Respondents’ General Knowledge Index (GKI), Personal Hygiene Index (PHI), and Temperature Knowledge Index (TKI)

The average of the General Knowledge Index (GKI) was 35.6 points (from a relative knowledge score of 79.1%) (Table 2). There was a statistically significant difference among the types of foodservice establishments, as determined by a one-way ANOVA analysis, Welch F (3698, *p* = 0.029). A Games–Howell post-test revealed that, compared to restaurants (35.84), cafes/bars and confectionery units had a statistically significant lower mean GKI score (33.45, *p* = 0.039). However, no statistically significant differences in means were found compared to small food establishments (35.48), which was also not confirmed in the Pichler et al. [11] study.

The GKI showed a slight negative correlation with education. Namely, as the level of education increases, the GKI score decreases. However, the one-way ANOVA did not show a statistically significant difference in index means between education levels. Those who did not receive any training had, on average, a statistically significantly lower score (33.6) compared to those who received training (35.67). Food handlers working in catering establishments are a frequent source of foodborne diseases (FBD) [5,12]; therefore, training and education is fundamental to food safety management [37]. According to EU food safety legislation [16], a food business operator should organise food safety training for employees. However, according to employees, the training provided by experts and work supervisors is the most effective means of ensuring proper food-handling practices [9].

Additionally, there are statistically significant differences (although these are very small) between persons who had, by 2004, completed the compulsory “Hygiene Minimum” standardised training course, and who achieved better scores on average (35.80) compared to those who did not (35.22). Another important factor seems to be the size of the establishment (measured by the average number of portions prepared on a usual day), where a small, negative, statistically significant correlation was observed.

The Personal Hygiene Index (PHI) average was 6.86 points (with a relative knowledge score of 85.8%). However, there were no statistically significant differences in means between the types of above-mentioned foodservice establishments (Welch F = 0.229, *p* = 0.796), something which was also not confirmed in the Pichler et al. study [12]. Minor and hard-to-explain differences were revealed for the PHI, and this suggested that men and smaller establishments fared slightly better. Knowledge of personal hygiene differs between men and women, with men achieving a slightly higher score on average (6.98) compared to women (6.74). In some other studies [38,39], no differences in food safety knowledge were noted based on gender. Again, a small, negative and statistically significant correlation was observed between the PHI and the size of the establishment.

The Temperature Knowledge Index (TKI), on the other hand, revealed poor knowledge (an average of 2.27 points, or 37.8%). However, the one-way ANOVA analysis (F = 4.288, *p* = 0.014) and Tukey post hoc test revealed a significant mean difference (*p* = 0.015) between restaurants (2.35), cafes/bars, and confectionery units (1.84) but not for the rest. TKI training seems to be an important factor, since those who did not receive any training averaged statistically significantly lower scores (1.80) compared to those who did (2.30). Additionally, those who completed the “Hygiene Minimum” training course achieved, on average, statistically significantly (albeit minimally) higher scores (2.34) compared to those who did not (2.16). This is consistent with various findings on the empowering impact that food safety training has on knowledge [11,21,36,37]. Pichler et al. [12] demonstrate a limited level of knowledge among food handlers in the catering industry concerning the optimal temperatures for cooking, holding, and storing foods.

For the general food safety knowledge index, we used a multiple linear regression analysis, which is presented in Table 3.

The GKI was set as a dependent variable in a regression analysis (Table 3). The independent variables included in the model were age, gender, being educated as a cook/waiter, completing food safety training and the “Hygiene Minimum” education, and food establishment size (measured as the no. of prepared meals). The model was statistically significant (F = 9.698, *p* < 0.001) and explains 9.1% of the variability in the GKI. Among the included independent variables, only gender (*p* = 0.611) did not affect the General Knowledge Index. Interestingly, age and establishment size negatively affected the GKI score, and, predictably, education as a cook or waiter, and having completed any corresponding and obligatory food safety training, positively affected food safety knowledge. However, we also observed a unique and statistically significant contribution of the “Hygiene Minimum” course. Namely, those who had completed this type of training course had, on average, about a 0.72 (on a scale from 0 to 45) higher score in food safety handling (while other independent variables were held constant).

### 3.3. Gaps in Food Safety Knowledge

Furthermore, we examined the most important gaps in food safety knowledge. Table 4 shows the frequency (N) and shares (%) of responses to critical items in the food safety field. We indicated as critical those items where the total share (%) of correct answers did not exceed 90%. It was observed that respondents were aware of microbiological risks, but still had some gaps in terms of food safety knowledge. One particular question with a noticeable lower percentage (68.3%) of correct answers was related to the following: “Is it true that if not completely cooked, raw beef could cause serious illnesses?” In a Pichler et al. study, 53% of respondents answered incorrectly [12].

Of all the respondents in our study, 79% falsely thought that food was safe when it smelled and tasted “normal”, which is significantly worse than the result (43%) from Pichler et al.’s study [12]. When asked about the temperature at which cold food should be stored, most food handlers (72.5%) chose the wrong answer (at 13 °C or lower). However, cross-contamination knowledge gaps were also revealed in response to statements such as the following: “Raw eggs may be stored above a prepared but uncovered salad in the refrigerator”, where 14.3% of respondents falsely thought this was true. For the statement that, “Beef may be placed directly on the counter to defrost”, 9.1% of respondents falsely thought this to be true. Finally, 13.1% falsely thought that “It is safe to put frozen chicken breast on the counter to thaw”. These gaps were previously identified in other studies [8,11,12]. Regarding the question, “Is it okay to put ice cubes in a glass by scooping the glass into the ice cubes?”, 14.6% choose yes, a false answer, which is similar to the study by Pichler et al. [12]. A total of 10% of respondents answered that often, sometimes, or rarely, they did not have time to wash their hands, even though they thought it would be necessary.

### 3.4. Limitations

An important aspect to be recognised is that we measured self-reported knowledge which, although it is a prerequisite, does not necessarily correspond to actual practices in food establishments. The authors, therefore, recommend an additional field observation study that would be able to uncover real-life behaviours and practices. It is also necessary to recognise that the COVID pandemic occurred after data collection, and this undoubtably had an affect on food establishments in general. We can speculate that, on one hand, awareness with regard to FBD has risen, but, on the other, after the layoffs which took place during the pandemic, the industry is now facing challenges in terms of recruiting adequate employees, and this might increase the risks associated with food-handling knowledge and skills. It would be therefore interesting to investigate if and how the situation has changed.

## 4. Conclusions

The results of the telephone survey showed that the interviewed food handlers did not have sufficient knowledge in the area of food safety (in particular, there was a lack of knowledge regarding refrigeration and cooking temperatures, and an inadequate knowledge of both pathogenic microorganisms and the measurement of food temperatures during cooking was noted). Of the three indexes composed for the purpose of this study, the Temperature Knowledge Index revealed the largest knowledge gaps, at a rate of 37.8%, compared to 85.8% for the Personal Hygiene Index, and 79.1% for the General Knowledge Index. Among the general and sociodemographic characteristics of the study’s participants, formal food-related education and on-the-job training had the greatest positive effect on the General Knowlege Index. Importantly, while holding the independent variables constant, the completion of the standardised “Hygiene Minimum” training course still contributed significantly to better food handling knowledge in general.

The results highlight the need to reconsider and possibly reorganise current food safety training approaches in Slovenia. Food handlers in food establishments are recognised as a risk factor if they are not properly educated and trained in food safety, as required by EU regulations. Therefore, we recommend periodic training by qualified food safety professionals and targeted training materials for different types of food establishments. If we do not provide proper training by qualified food safety professionals, we cannot expect to have a proper food safety culture that stipulates how professional food-handling employees should work according to food safety requirements.

## Figures and Tables

**Table 1 foods-12-02735-t001:** The sample characteristics of respondents.

Variable	Category	N	% *
Gender (N = 601)	Male	308	51.3
	Female	293	48.8
Age (N = 587)	16–30	84	14.3
	31–45	258	44.0
	46–60	204	34.8
	61–75	41	7.0
Education (N = 600)	Basic education	25	4.2
	Secondary technical education (cook/bartender)	248	41.3
	Secondary technical education (other)	170	28.3
	General upper or secondary level education	39	6.5
	Post-secondary and higher education	118	19.7
Country of birth (N = 600)	Slovenia	493	82.2
	Other	107	17.8
Native language (N = 599)	Slovenian	516	86.1
	Other	83	13.9
Attended any form of food safety training (N = 601)	Yes	566	94.2
	No	35	5.8
Attended the “Hygiene Minimum” training course (N = 601)	Yes	345	57.4
	No	256	42.6
Years of work experience as a food handler (N = 601)	1–10	231	38.4
	11–20	153	25.5
	21–30	129	21.5
	31–40	72	12.0
	41–50	16	2.7
Position of the employee	Manager	235	39.1
(N = 601)	Chef	206	34.3
	Assistant chef	73	12.1
	Kitchen assistant	62	10.3
	Other	25	4.2
Type of establishment (N = 601)	Restaurants and canteens	364	60.6
	Bars, cafes, and confectioneries	38	6.3
	Snack bars	122	20.3
	Institutional food providers	2	0.3
	Other	75	12.5

* Percentages may not sum up to 100 due to rounding. In cases where the sample size was not 601, this was due to item non-response, which was most often incurred by explicit refusal, or “I don’t know,” responses.

**Table 2 foods-12-02735-t002:** A description of variables regarding the General Knowledge Index (GKI), Personal Hygiene Index (PHI), and Temperature Knowledge Index (TKI).

Variables/Indices	Statistic	General Knowledge Index (GKI) (0–45)	Personal Hygiene Index (PHI) (0–8)	Temperature Knowledge Index (TKI) (0–6)
	Min	10	3	0
	Max	42	8	5
	M	35.60	6.86	2.27
	SD	3.10	0.90	1.08
Gender	t (p)	0.877 (0.381)	**3.401 (0.001)**	1.056 (0.292)
Age	r (p)	−0.012 (0.766)	0.006 (0.890)	−0.028 (0.498)
Years of experience	r (p)	0.045 (0.268)	−0.064 (0.114)	0.037 (0.367)
Education	r_s_ (p)	**−0.081 (0.048) ***	−0.051 (0.216) *	−0.044 (0.282)
Food safety training	t (p)	**3887 (<0.001)**	1602 (0.110)	**2638 (0.009)**
Hygiene minimum training	t (p)	**−2.265 (0.024)**	0.823 (0.411)	**−1.997 (0.046)**
Native language (Slovenian/other)	t (p)	1649 (0.100)	0.066 (0.948)	1.862 (0.063)
Size (No. of meals)	r (p)	**−0.222 (<0.001)**	**−0.170 (<0.001)**	−0.074 (0.069)

Notes: Min = Minimum; Max = Maximum; M = Mean; SD = Standard Deviation; t = t-statistic, r = Pearson correlation coefficient, r_s_ = Spearman’s correlation coefficient, p-probability value. * The one-way ANOVA did not show a statistically significant difference in index means between education levels.

**Table 3 foods-12-02735-t003:** A multiple linear regression analysis for the General Knowledge Index.

Variables	B	SE(B)	β	*p*
(Constant)	35,275	0.748		<0.001
Age	−0.030	0.014	−0.111	0.030
Gender (male)	−0.131	0.258	−0.021	0.611
Education (cook/waiter)	0.517	0.261	0.082	0.048
Food safety training (yes)	1639	0.538	0.123	0.002
Hygiene Minimum training (yes)	0.720	0.316	0.116	0.023
Establishment size (no. of meals)	−0.004	0.001	−0.244	<0.001

Notes: F (6, 580) = 9.698 *p* < 0.001, R^2^ = 0.091.

**Table 4 foods-12-02735-t004:** The frequency (N) and shares (%) of responses to critical items in the food safety field.

Questions	Respondents’ Answers (Valid N, %)	
	True N (%)	False N (%)
Is it true that if not completely cooked, these foods could cause serious illnesses? Raw beef (true)	411 (68.4) ✓	190 (31.6)
It is safe to put frozen chicken breast on the counter to thaw? (false)	79 (13.1)	522 (86.9)
You can be sure food is safe to eat when it smells and tastes normal. (false)	476 (79.2)	125 (20.8) ✓
Cold food should be stored chilled (at 13 °C or lower). (false)	436 (72.5)	165 (27.5) ✓
Raw eggs may be stored above a prepared but uncovered salad in the refrigerator. (false)	86 (14.3)	515 (85.7) ✓
The chilling of hot food has to happen quickly. To pass through the critical temperature range faster, it is recommended to put the food into smaller containers for storage in the refrigerator. (true)	542 (90.2) ✓	59 (9.8)
Beef may be placed directly on the counter to defrost. (false)	55 (9.1)	546 (90.9) ✓
Raw meat can be stored anywhere in a refrigerator if it is tightly sealed in plastic film. (true)	151 (25.1) ✓	450 (74.9)
Properly labelled detergents may be kept in the same areas where food is prepared if they have their own storage area and are only used for intermediate cleaning. (true)	262 (43.6) ✓	339 (56.4)
Is it okay to put ice cubes in a glass by scooping the glass into the ice cubes? (no)	Yes 88 (14.6)	No 513 (85.4) ✓
Do you need to have thoroughly washed hands if you use a food processor or gripper when handling food? (yes)	Yes 521 (86.7) ✓	No 80 (13.3)
Do you ever not have time to wash your hands, even though you think it would be necessary? (never)	Never 541 (90.0) ✓	Often, sometimes, rarely 60 (10.0)

Notes: The variables have been recoded, and the answer, “wrong,” also includes the answer “I don’t know,” from the original variables. Beside the indication of a question/statement, the symbol “✓” indicates a correct answer, and a highlighted space (grey) indicates a wrong answer.

## Data Availability

The data presented in this study are available on request from the corresponding author.

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
