# Peer review of "Food Safety Knowledge among Professional Food Handlers in Slovenia: The Results of a Nation-Wide Survey"

_foods, 2023, doi:10.3390/foods12142735_

Round 1

Reviewer 1 Report

Line 165. Don’t start a sentence with a number unless spelled out, suggest using a semicolon between the two sentences. “…their mother tongue; 38.4% of respondents had 1-10…”

Table 1. What does gymnasium signify?

Line 238. Presumably thawing instead of melting (chicken breasts).

Table 2. Any table should stand alone for interpretation, therefore there has to be an explanation for the letters in the column under Statistics, i.e., t (p), r(p), etc.

Line 350. Check whether Covid or COVID (or COVID-19). I have been using COVID.

Author Response

The authors wish to thank the reviewers for their valuable time to comment on the manuscript in such a constructive way.

Regarding comments raised after our submission, please find below list of changes made. Our changes are visible in the revised version of the manuscript where deleted text and text which was moved to another location inside the manuscript are visible as track changes. Text that was changed or added is visible in red colour.

 Line 165. Don’t start a sentence with a number unless spelled out, suggest using a semicolon between the two sentences. “…their mother tongue; 38.4% of respondents had 1-10…”

Authors: We have corrected the sentence.

 Line 238. Presumably thawing instead of melting (chicken breasts).

Authors: Corrected.

Table 2. Any table should stand alone for interpretation, therefore there has to be an explanation for the letters in the column under Statistics, i.e., t (p), r(p), etc.

Authors: Explanations added.

  Line 350. Check whether Covid or COVID (or COVID-19). I have been using COVID.

Authors: We changed the word Covid to COVID.

Reviewer 2 Report

In this manuscript entitled "Food safety knowledge among professional food handlers in Slovenia: results of a nation-wide survey", the authors have studied the KAP of food handlers in Slovenia. The following issues are to be addressed carefully:

Ø  Line 103: There are a number of probability sampling designs, such as simple random sampling, systematic sampling, stratified sampling, cluster sampling, and multi-stage sampling. So please be specific about the approach used.

Ø  Line 105: It should be 4300.

Ø  Lines 110–112: "He then wrote down their names and functions, and finally the computer randomly selected a person to participate in the survey". Rephrase. How was the computer randomly selected? What tool was used? Please be specific and scientific.

Ø  Line 121: Instead of "pilot tested", use the term content validation."

Ø  Line 122: Cronbach’s alpha test is usually used for testing the reliability and suitability of a questionnaire. Have the authors used. Discuss

Ø  Section 2.2: Are the terminologies "temperature knowledge index (TKI)", "personal hygiene index (PHI)" and "general knowledge index (GKI)" coined by the authors or adapted from any previous studies? Please cite a suitable reference. If not, I feel these indices are misleading in the context of food safety.

Ø  Line 146: "The general knowledge index (GKI) measures knowledge of food handling and general hygiene in the kitchen." Does it make sense?

Ø  Section 2.3: "Chi-square test, logistic regression, one-way ANOVA, independent samples t-test, Pearson and Spearman correlation coefficient" Have you used all these tests? Please be specific about what test was used for what purpose.

Ø  Section 3.1: "Sample characteristics". Rename it "Demographic characteristics of respondents."

Ø  Table 4: Total the correct scores with the mean and standard deviation.

Ø  3.4. Limitations: Be sure to point out that it's possible that the methods used here don't represent how things really work in commercial kitchens.

Ø  Discussion: Discussion is written poorly. Kindly rewrite, highlighting important results.

Ø  Conclusions: This is a very lengthy conclusion; shorten it. Just mention the food handlers with good levels of knowledge (mean±SD), attitudes, and practises, and discuss any positive association among levels of knowledge. 

 Extensive editing of English language required

Author Response

The authors wish to thank all reviewers for their valuable time to comment on the manuscript in such a constructive way.

Regarding comments raised after our submission, please find below list of changes made. Our changes are visible in the revised version of the manuscript where deleted text and text which was moved to another location inside the manuscript are visible as track changes. Text that was changed or added is visible in red colour.

In this manuscript entitled "Food safety knowledge among professional food handlers in Slovenia: results of a nation-wide survey", the authors have studied the KAP of food handlers in Slovenia. The following issues are to be addressed carefully:

Ø  Line 103: There are a number of probability sampling designs, such as simple random sampling, systematic sampling, stratified sampling, cluster sampling, and multi-stage sampling. So please be specific about the approach used.

Authors: Thank you for your comment. Indeed, we now state more clearly that simple random sampling was used to select respondents at food establishment level. The exact procedure is also described in detail in section. 2.1.

Ø  Line 105: It should be 4300.

Authors: Corrected.

Ø  Lines 110–112: "He then wrote down their names and functions, and finally the computer randomly selected a person to participate in the survey". Rephrase. How was the computer randomly selected? What tool was used? Please be specific and scientific.

Authors: We have rephrased the text for clarity. However, we do not know the exact name of the application that the reputable professional research agency contracted for data collection used for random selection of the participants. We can provide this information at a later date if necessary.

Ø  Line 121: Instead of "pilot tested", use the term content validation."

Authors: Corrected.

Ø  Line 122: Cronbach’s alpha test is usually used for testing the reliability and suitability of a questionnaire. Have the authors used. Discuss

Authors: As mentioned in section 2.2 the questionnaire was developed by the University of Illinois at Chicago School of Public Health, it has been used, and improved several times (see Pichler et al. [11], Dworkin et al. [35] and Panchal et al. [36]), moreover, our Slovenian version was also content validated, and pilot tested. Therefore, we are confident in suitability, validity, and reliability of the questionnaire. Regarding Cronbach’s alpha as a measure of reliability (internal consistency), we would like to point out that the questionnaire measures knowledge. So do the constructed indexes, which are composite variables. Since these are formative constructs, multicollinearity between items is not desired and Cronbach’s alpha is not an appropriate measure (see, e.g., Hair et. al., 2019).

Ø  Section 2.2: Are the terminologies "temperature knowledge index (TKI)", "personal hygiene index (PHI)" and "general knowledge index (GKI)" coined by the authors or adapted from any previous studies? Please cite a suitable reference. If not, I feel these indices are misleading in the context of food safety.

Authors: The authors coined the terms to refer to specific areas of knowledge related to food safety. The language editor has now been consulted for its appropriateness and have suggested that the terms remain.

Ø  Line 146: "The general knowledge index (GKI) measures knowledge of food handling and general hygiene in the kitchen." Does it make sense?

Authors: Slightly rephrased for clarity.

Ø  Section 2.3: "Chi-square test, logistic regression, one-way ANOVA, independent samples t-test, Pearson and Spearman correlation coefficient" Have you used all these tests? Please be specific about what test was used for what purpose.

Authors: Yes, all tests indicated were used for bivariate analyses. Listing the tests for all individual analyses would take up a lot of space, and we believe that the use of an appropriate test is also evident from the precise reporting of results in the tables and text.

Ø  Section 3.1: "Sample characteristics". Rename it "Demographic characteristics of respondents."

Authors: Corrected.

Ø  Table 4: Total the correct scores with the mean and standard deviation.

Authors: The purpose of Table 4 is to show the larges gaps in knowledge by presenting the shares of incorrect answers to specific questions. Since these data are nominal, mean values and standard deviations cannot be calculated. However, these are presented for TKI, PHI, GKI indexes (see Table 2.)

Ø  3.4. Limitations: Be sure to point out that it's possible that the methods used here don't represent how things really work in commercial kitchens.

Authors: We think that the first two sentences express exactly these limitations of the study.

Ø  Discussion: Discussion is written poorly. Kindly rewrite, highlighting important results.

Authors: According to the journal instructions for authors that the discussion part can be combined with the results, we chose this kind of approach to discuss and explain the results in more detail to the reader.

In shortening the conclusions (based on the following commentary), we have found that part of the text written in the conclusion would be therefore better included in the discussion. Some of the text that has been taken from the conclusions and meaningful inserted into the discussion is shown in red for ease of reference.

Ø  Conclusions: This is a very lengthy conclusion; shorten it. Just mention the food handlers with good levels of knowledge (mean±SD), attitudes, and practices, and discuss any positive association among levels of knowledge. 

Authors: We have shortened the conclusions as suggested. To avoid opacity due to the use of "track changes," we have rewritten it. However, we would like to point out that the aim of this study was to identify the main knowledge gaps in the field of food safety. For this reason, we focus only on knowledge, not attitudes and practices, in the conclusions. The levels of knowledge with the biggest gaps and positive associations are highlighted through the three indexes composed for the purpose of this study.

Extensive editing of English language required.

            Authors: The manuscript has been carefully checked for English language. All corrections are marked as track changes.

Reviewer 3 Report

This is an excellent well conducted and written study, with an original contribution to the knowledge of food safety among professional food handlers in Slovenia: results of a nation-wide survey.

I encourage  minor modifications as outlined below:

L25: The Introduction section is very long. Please try to short this section and be more concise.

L26: “more than half (57.5%) of adult population” please give some numbers. This recommendation is regarded for better view of the current situation.

L29: “considered as an important source of foodborne diseases” please give some examples of foodborne diseases.

L102: “we designed” - personal style is not recommended in scientific manuscripts. Please try to rephrase.

L105: “601 agreed to take part” people or businesses? Please mention this aspect in the manuscript.

L160: At the “Results and discussion” section you can add one or two phrases about One Health concept and it’s status in Slovenia.

L224: I’m wondering it’s “recoded” or recorded.

L356: The “Conclusion” section is too long. Please try to find a more easy to read for the reader and specific ideas for this section.

Author Response

The authors wish to thank all reviewers for their valuable time to comment on the manuscript in such a constructive way.

Regarding comments raised after our submission, please find below list of changes made. Our changes are visible in the revised version of the manuscript where deleted text and text which was moved to another location inside the manuscript are visible as track changes. Text that was changed or added is visible in red colour.

This is an excellent well conducted and written study, with an original contribution to the knowledge of food safety among professional food handlers in Slovenia: results of a nation-wide survey.

I encourage minor modifications as outlined below:

L25: The Introduction section is very long. Please try to short this section and be more concise.

Authors: We have shortened the introduction where possible without compromising the overall presentation of the substantive framework that was the reason for planning and conducting the survey. Some sections have been rewritten. Changes made in the text are visible as »track changes«.

L26: “more than half (57.5%) of adult population” please give some numbers. This recommendation is regarded for better view of the current situation.

            Authors: Explanations added.

L29: “considered as an important source of foodborne diseases” please give some examples of foodborne diseases.

Authors: Explanations added.

L102: “we designed” - personal style is not recommended in scientific manuscripts. Please try to rephrase.

            Authors: We have used passives now.

L105: “601 agreed to take part” people or businesses? Please mention this aspect in the manuscript.

Authors: Explanations added.

L160: At the “Results and discussion” section you can add one or two phrases about One Health concept and it’s status in Slovenia.

            Authors: We prepared an explanation about the concept of One Health in Slovenia, but somehow it doesn't fit in the Results and discussion section.

in Slovenia, within the "One Health" concept, two ministries are responsible (the Ministry of Health and the Ministry of Agriculture, Forestry and Food), which annually implement the Program for Monitoring Zoonoses and Their Agents, and monitoring the resistance of isolates to antimicrobial drugs. The national strategy "One Health" for managing microbial resistance (2019-2024) was also adopted. Public health services and veterinary services have realized the use of the One Health approach in the exchange of information in case of suspected or confirmed infection with the novel coronavirus in animals - to carry out a risk assessment when a person who has contracted COVID-19 has been in contact with pets or other animals.

The "One health" approach (health of people, animals and ecosystems) also includes the cooperation of professional services and inspectors in preventing the introduction and spread of harmful plants and the work of individuals in maintaining healthy plants.

L224: I’m wondering it’s “recoded” or recorded.

            Authors: We have rewritten the sentence.

L356: The “Conclusion” section is too long. Please try to find a more easy to read for the reader and specific ideas for this section.

Authors: We have shortened the conclusions as suggested and highlighted important results.